# Predicting road quality using high resolution satellite imagery: A transfer learning approach

**Ethan Brewer**[1]*, **Jason Lin**[1], **Peter Kemper**[2], **John Hennin**[1], **Dan Runfola**[1]

**1** Department of Applied Science, William & Mary, Williamsburg, VA, United States of America, **2** Department of Computer Science, William & Mary, Williamsburg, VA, United States of America

* embrewer@email.wm.edu

**Data Availability Statement:** Raw road data and Android app code is available at https://github.com/wmgeolab/roadrunner_app.

**Funding:** This work was supported in part by the Commonwealth Cyber Initiative (CCI), an

## Abstract

Recognizing the importance of road infrastructure to promote human health and economic development, actors around the globe are regularly investing in both new roads and road improvements. However, in many contexts there is a sparsity—or complete lack—of accurate information regarding existing road infrastructure, challenging the effective identification of where investments should be made. Previous literature has focused on overcoming this gap through the use of satellite imagery to detect and map roads. In this piece, we extend this literature by leveraging satellite imagery to estimate road quality and concomitant information about travel speed. We adopt a transfer learning approach in which a convolutional neural network architecture is first trained on data collected in the United States (where data is readily available), and then "fine-tuned" on an independent, smaller dataset collected from Nigeria. We test and compare eight different convolutional neural network architectures using a dataset of 53,686 images of 2,400 kilometers of roads in the United States, in which each road segment is measured as "low", "middle", or "high" quality using an open, cellphone-based measuring platform. Using satellite imagery to estimate these classes, we achieve an accuracy of 80.0%, with 99.4% of predictions falling within the actual or an adjacent class. The highest performing base model was applied to a preliminary case study in Nigeria, using a dataset of 1,000 images of paved and unpaved roads. By tailoring our US-model on the basis of this Nigeria-specific data, we were able to achieve an accuracy of 94.0% in predicting the quality of Nigerian roads. A continuous case estimate also showed the ability, on average, to predict road quality to within 0.32 on a 0 to 3 scale (with higher values indicating higher levels of quality).

## Introduction

Investments in road infrastructure are a major expenditure of both international development organizations and local governments, reflecting the importance of transportation networks for a wide range of human outcomes [1–16]. Despite the importance of road networks, data on their location and quality is sparse in much of the world, particularly in developing nations [17]. While a selection of recent research (i.e., [18]) has sought to identify where roads are located using satellite data, a much smaller body of literature has explored the topic of road

investment in the advancement of cyber R&D, innovation, and workforce development. For more information about the CCI, visit www.cyberinitiative.org. The funder had no role in study design, data collection and analysis, decision to publish, or preparation of the manuscript.

**Competing interests:** The authors have declared that no competing interests exist.

quality (i.e., [17]). This challenges our ability to effectively allocate resources, as without accurate measures of road quality—and concomitant measures of the speed of travel—it is difficult to estimate the impact a new or improved road may have on key metrics, such as travel times to local markets or clinics. Approaches employed for the measurement of road quality to date have had critical limitations, with local measurements requiring large amounts of time, labor, and expensive equipment [17], and crowdsourced information being plagued by sparse collection and inaccuracy in many of the locales where data is needed most [19, 20].

Predicting road quality using remote, automated analysis of high resolution images collected from aerial or satellite platforms provide a globally systematic solution to this challenge [21]. In this paper, we present a test of the use of transfer learning in convolutional neural networks in conjunction with high resolution satellite imagery of roads to determine (a) if road quality can be estimated with a reasonable degree of accuracy with satellite imagery, and (b) the degree to which such an approach can be applied across different geographies.

Our paper is structured as follows. In "Related works", we review the relevant literature on the application of computer vision to satellite imagery. In "Data", we discuss our data collection methods, and "Methodology" provides the technical approach we test for road quality classification. We introduce our results in the following section, and finally provide a discussion and conclusion in the final two sections.

## Related works

Recent improvements in the quality and speed of Convolutional Neural Networks (CNNs) has led to several novel applications in many domains including for satellite imagery [22–27]. One of the most prominent examples of this has been recent research into the capability of daytime satellite imagery to predict factors traditionally only collected with on-the-ground surveys, including household income and factors related to health outcomes [28–31]. Progress on identifying the limits and opportunities of satellite sources has been swift, with the computer vision and remote sensing (RS) communities collaborating to overcome a number of challenges. A wide range of literature has provided insights into effective technical strategies to overcome these differences; [22] and [32] provide a broad overview of the technical objectives and innovations that have emerged over the last few years; we further provide our own review in S1 Appendix.

Specific to roadways, research has been conducted on road detection, centerline extraction, mapping road safety, and automated road crack detection [18, 33–35]. Remote sensing road detection literature has a long history, going back to efforts in the 1980s and earlier using coarse resolution imagery and manual digitization [36]. As with other image analyses, the difficulties of road detection from remotely sensed images lie in that the image characteristics of road features can be affected by sensor type, spectral and spatial resolution, weather, light variation, and ground characteristics, among other factors [36]. Additionally, given RS images of roads often contain discontinuities, occlusion or shadows, near-parallel boundaries with constancy in width, and sharp bends, it is difficult to model all these situations and to incorporate them into a single module—in practice, a road network is too complex to be modeled using a general structural model [36]. Most of the methods suggested in literature for road detection consist of one or more types of algorithms: classification-based (NNs and SVMs) [37, 38], knowledge-based [39, 40], mathematical morphology [41, 42], active contour model [43], and dynamic programming [44]. In more recent times, convolutional networks have begun to be tested for their efficiency at road network detection and extraction. Zhang et al. merged GF-2 and World View satellite images as the input for a CNN to extract roads, and achieved an accuracy of 99.2% [45]. Similarly, Xu, Mu, Zhao, and Ma, used low and high frequency sub-bands

that reflect multiscale image features which were obtained by a contourlet transform, obtaining a scene classification accuracy above 90% [46]. Furthermore, Xia, Cao, Wang, and Shang added four types of texture information to satellite images and used the resulting data as the input for a CNN to extract roads, vehicles, and vegetation based on the CNN and conditional random field methods [47]. These studies helped illustrate that greater texture and spectrum information in multisource data can improve the accuracy of extracting road information from RS images [22].

A far more limited literature has explored our capability to discern road quality with satellite imagery. In 2018, Oshri et al. used two data sources in a supervised learning setting: survey data from Afrobarometer [48] as ground truth infrastructure quality labels, and relatively low resolution (10 and 30 meter) satellite imagery from Landsat 8 and Sentinel 1 as input sources to classify, among other infrastructure items, road quality in a binary fashion [49]. They achieved 70.5% accuracy. Also in 2018, Gabriel Cadamuro et al. carried out work utilizing CNNs to classify road quality from satellite images [17]. Here, the International Roughness Index (IRI) was collected by specialized equipment for over 7,000 km of predominantly trunk roadways in Kenya. Using the IRI information to label 50x50 cm resolution satellite images of the corresponding roads, the pre-trained networks AlexNet, VGG, and SqueezeNet were used to classify road segments, yielding accuracy scores shown in Table 1 for binary and 5-category classification. Cadamuro et al. described two key remaining challenges to road classification: (1) treating the problem as sequential, i.e., for the prediction of a given road segment, utilize the data of nearby road segments, and (2) better accommodation for the continuous nature of road roughness measurements to mitigate the negative impact of road heterogeneity on the quality of predictions [17]. This paper contributes to this growing body of literature in a number of ways, including tests incorporating higher resolution imagery, exploration of the effectiveness of intercontinental transfer learning, and the implementation of a continuous measurement of road roughness.

## Data

This section details our data collection and labeling strategy. A total of 53,686 images of roads in Virginia were collected and labeled according to their quality following the process detailed in this section. Additionally, 1,000 images of roads throughout Nigeria were used to test the transferability of the best identified model architecture.

### Road roughness collection

In order to label the satellite imagery, road roughness values were collected via an Android app. The app (source code at https://github.com/wmgeolab/roadrunner_app) was distributed

**Table 1. Cadamuro et al. results.**

| Architecture | Binary | | 5-class | |
|---|---|---|---|---|
| | Standard (%) | Held-out (%) | Standard (%) | Held-out (%) |
| SqueezeNet (64) | 88 | 79 | 73 | 52 |
| SqueezeNet (224) | 89 | 84 | 69 | 49 |
| VGG-11 (64) | 90 | 79 | 71 | 51 |
| VGG-11 (224) | 87 | 78 | 65 | 44 |
| AlexNet (64) | 89 | 79 | 70 | 52 |
| AlexNet (224) | 87 | 79 | 64 | 45 |

Cadamuro et al.'s accuracy results for binary and 5-class prediction of road quality. "Standard" was a random train-test split and "held-out" was a train-test split where roads within their 1 km block were not included in both the train and test sets. Batch sizes are in parentheses.

to a total of thirty trained users for collection throughout Virginia, USA from September to November 2020. Information was collected across 2,400 km of primary, secondary, and unpaved roads in southeastern, central, northern, and western Virginia, USA.

The data collection application uses a device's accelerometers to measure the average movement of the device, with respect to the vehicle, over a 20–70 meter driving distance. This movement is essentially vibratory information transmitted through the tires, suspension, and ultimately to whatever part of the vehicle the device is in contact with. We will refer to this measurement as the road segment's average vibration, $\bar{v}$. The app records $\bar{v}$ in three directions with respect to the device screen. The app records only when driving over a minimum speed, such as 3 mph. Geographic coordinates (latitude and longitude) are recorded whenever new pairs become available through the device's GPS manager. Our version of the app builds on the previous correlation identified between phone metrics and road roughness established in the software created by Mark Buie (https://github.com/mebuie).

Due to the variable nature of vehicle ride and the placement and physical properties of a given device, the relationship between $\bar{v}$ and road quality was individually determined for each device/vehicle setup. For each setup, subjective visual and somatosensory judgements of road quality were made by the user as they drove and collected data, noting the time of collection as well. For our approximations, relatively smooth, maintained, highway and primary roads are deemed high quality. Low quality roads are unpaved dirt or gravel roads, or road segments inflicted with numerous potholes. Mid-quality roads fall somewhere in-between, including roads with grainy textures like concrete or road segments with some potholes present. Using this judgement along with time of collection and the corresponding average vibration, the qualities of road segments were inferred. See Fig 1 as an example of how the average vibration values were used to sort the quality of each segment for a particular user.

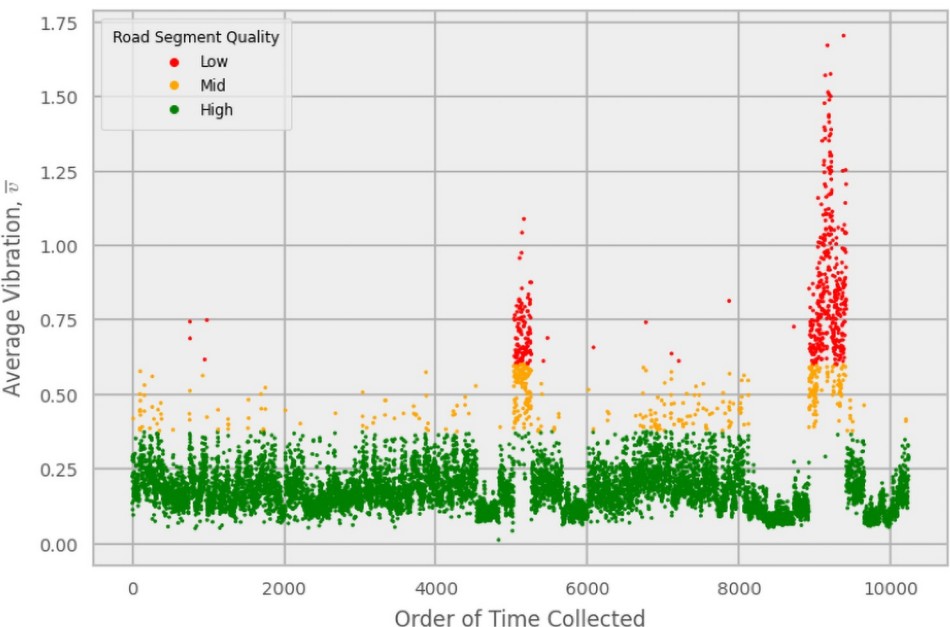

**Fig 1. Average vibrations.** Plot of the average vibrations, $\bar{v}$, for an individual device in the order of time collected, corresponding to dates between September 4th and November 26th, 2020. In this case, $\bar{v}$ in the direction of the center-of-Earth to sky axis was used. Each point represents a road segment. The quality thresholds for this setup were determined to be 0.375 and 0.600, where points falling below $\bar{v} = 0.375$ represent high quality roads, those above $\bar{v} = 0.600$ are low quality, and points in-between are mid-quality. The spike on the right corresponds to a particular drive on dirt roads in the Appalachians near Blacksburg, VA, USA.

## Remotely sensed imagery collection

The source images were supplied by the Virginia Geographic Information Network's Virginia Base Map Project (VBMP) with 30x30 centimeter resolution pixels captured between 2017 and 2019 [50]. As needed, region images were manually downloaded and merged into a single composite (no overlaps were present in the collection grid). For the duration of data collection, a processing script was run continuously in cycles to analyze new road segment entries in near-real-time from the cellphone collection devices. The processing script took the geographic coordinates of a given segment and used them to crop a new image showing only the imagery contained within the minimum and maximum latitudes and longitudes, with 5 pixels added to each as buffer. The result was a collection of images, with one image corresponding to each road segment traversed (see Fig 2).

Of the 53,686 collected road segments, 47,557 were labeled high-quality roads, 5,417 were mid-quality, and 712 were low quality. This imbalance was taken into account when creating the testing set by composing approximately half the testing set with high-quality roads, a quarter with mid-quality, and a quarter with low quality. This strategy allowed for a more stringent test of our ability to detect low and mid-quality quality roads, while still seeking to minimize balance errors.

The road quality information for the test set of Virginia-independent roads was obtained from the Africa Infrastructure Country Diagnostic (AICD), a comprehensive knowledge program commissioned by the World Bank and the Infrastructure Consortium for Africa to improve understanding of Africa's infrastructure [51]. Data was collected from 2001–2006. The data of interest consisted of geographic coordinates for roads, within Nigeria in our case, and their associated pavement type and condition. Centered around a given coordinate pair, Google Maps Static API was used to clip corresponding 640x640 pixel images of the road

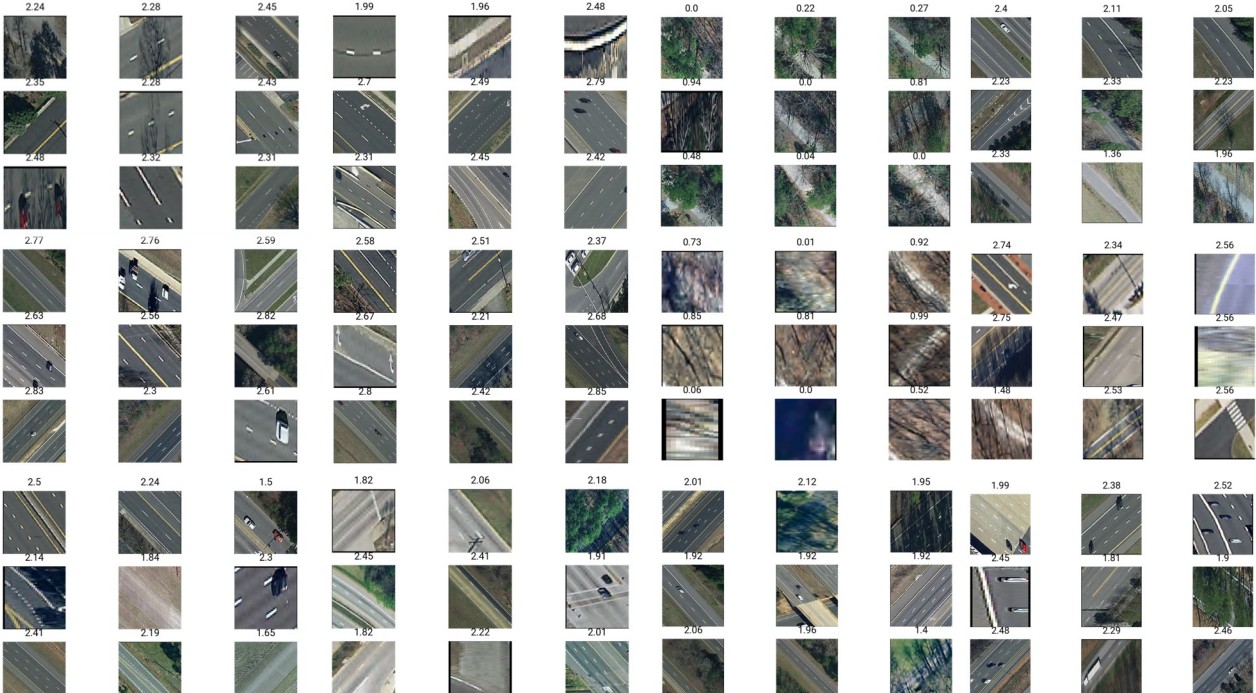

**Fig 2. Random Virginia images.** Random sample of cropped road segment images resized to 200x200 with corresponding labels. Values in the range [0–1] correspond to low quality roads, [1–2] corresponds to mid-quality, and [2–3] for high quality.

segments [52]. For our study, unpaved dirt and gravel roads were labeled low quality and paved roads were labeled a higher quality, for a total of two classes (enabling comparison to the prior literature in [17]). The roads were selected from diverse geographic locations throughout the country. Of the 1,000 total Nigerian roads selected, 500 were unpaved and 500 were paved. Pavement type was confirmed by visual inspection of the individual images. On average, the road segments in the Nigerian images comprised less of the image space than those in Virginia due to differences in cropping methods. It is important to note most of the unpaved roads in Nigeria have a distinctive orange color due to soil composition in the region.

## Methodology

The overall methodology for this project was the following:

1. Collect vibration data on road segments from the cabin of vehicles

2. Analyze this data to represent three levels of road roughness: high quality, mid-quality, and low quality

3. Use this information to label corresponding satellite image crops of those road segments

4. Contrast the capability of a range of CNN architectures to accurately classify road segments

5. Test the networks on a subset of Virginia images not used in training

6. Test the transfer learning potential of this model with Nigerian roads

We focus on testing a range of individual model architectures common in the literature, as well as stacked generalization ensembles. The specific models that are contrasted in this piece to assess their capabilities in discriminating between road classes included ResNet50 [53], ResNet152V2 [53], Inceptionv3 [54], VGG16 [55], DenseNet201 [56], InceptionResNetV2 [57], and Xception [58]. Each model was pre-trained on ImageNet; a version of ResNet50 pre-trained on BigEarthNet was also tested.

For each base model architecture, data preparation included resizing images to 200x200 pixels using bilinear interpolation for standardized input into the pre-trained architectures. A total of 52,821 training and validation images were used with a 75/25% split (N = 39,616 for training; N = 13,205 for validation). Additionally, a test set of 865 was withheld for later testing, and not used during the fitting process.

For each image, a scaled continuous value—Continuous Quality Value (*CQV*)—of road quality in the range [0–3] was derived using a linear model based on the measured average vibrations and the associated threshold values between classes for each cellphone device. For each device setup, the *CQV* function was a combination of three linear lines (see Fig 3 for an example). The upper limit for a high quality road would be a perfectly smooth road having a $\bar{v}$ of 0.0, corresponding to a *CQV* of 3.0. The lower limit $\bar{v}$ of high quality roads is defined, by definition, to be the average vibration between the high and mid classes, $\bar{v}_{hm}$, with a *CQV* of 2.0. The slope of the mid-quality portion of the function is bounded by $\bar{v}_{hm}$ and the mid/low threshold, $\bar{v}_{ml}$, with *CQV*s between 2.0 and 1.0. The upper limit for the low quality roads corresponding to *CQV* = 1.0 is, by definition, $\bar{v}_{ml}$, with the slope of the line equal to the average of the slopes of the other two segments. In all,

$$CQV(\bar{v}) = \begin{cases} 3 - (\bar{v}/\bar{v}_{hm}) & \bar{v} < \bar{v}_{hm} \\ 2 - (\bar{v} - \bar{v}_{hm})/(\bar{v}_{ml} - \bar{v}_{hm}) & \bar{v}_{ml} > \bar{v} \geq \bar{v}_{hm} \\ 3 - (2\bar{v}/\bar{v}_{ml}) & \bar{v} \geq \bar{v}_{ml} \end{cases} \qquad (1)$$

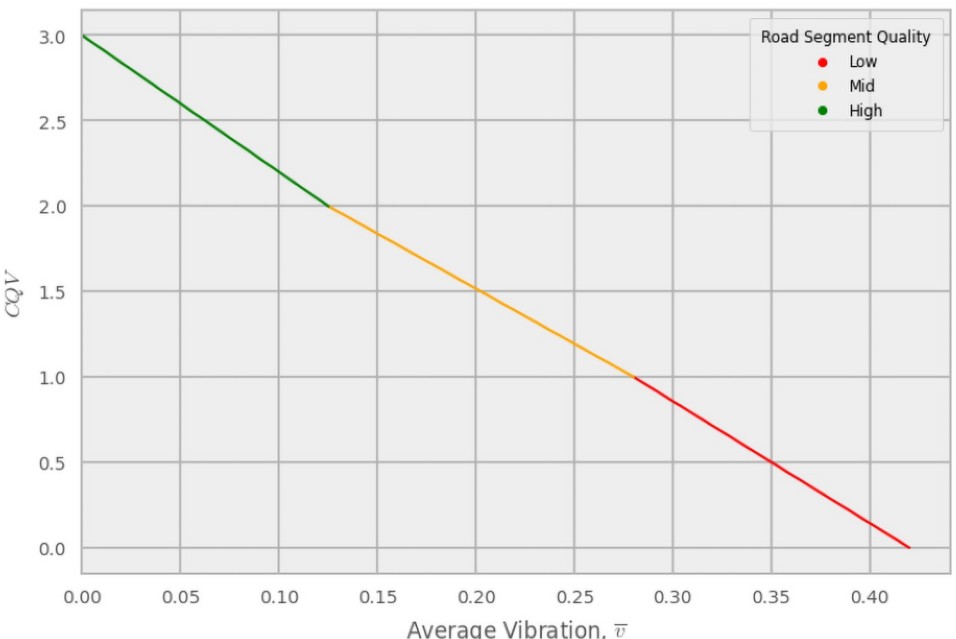

**Fig 3. Continuous quality value.** Example of the determination of the continuous quality value (*CQV*) of a road segment for a particular device setup. The line is split into 3 parts, representing each of the three quality classes. The slope for the [2–3] quality range is based on the *CQV* = 2.0 threshold value for the device setup and the constraint of the upper range, *CQV* = 3.0. The slope for the [1–2] range is determined from the *CQV* = 1.0 and *CQV* = 2.0 thresholds. Finally, the slope of the [0–1] range is the average of the combined range [1–3]. Negative *CQV*s (i.e., roughness values greater than the maximum defined) were re-specified to 0. In this example, the threshold between high and mid-quality roads, $\bar{v}_{hm}$, is 0.125, and the threshold between mid and low, $\bar{v}_{ml}$, is 0.28. The colors red, orange, and green correspond to the regions of low, mid, and high quality roads, respectively.

where, again, $\bar{v}$ is the average vibration of a given road segment and $\bar{v}_{hm}$ and $\bar{v}_{ml}$ are the average vibration thresholds between high and mid-quality roads and mid and low quality roads, respectively, for a given device setup.

Because of inherent uncertainty in the human perception of road quality and unavoidable non-road induced vibrations present in this study, a form of label smoothing was calculated and used for training to provide a more nuanced representation of the data. The goal of this smoothing is to ascribe a probability a given sample belongs in the "low", "mid", or "high" quality ranges, rather than only provide a single classification. This was accomplished for each road segment by placing its *CQV* as the mean in a normal distribution and computing the areas under the curve bounded by each of the three classes. That is,

$$P_i = \int_a^b \frac{e^{\frac{1}{2}\left(\frac{CQV - CQV_0}{\sigma}\right)^2}}{\sigma\sqrt{2\pi}} \, dCQV \tag{2}$$

where $CQV_0$ is the *CQV* of a given road segment, $\sigma$ is the standard deviation (equal to 0.25 in this case), and $P_i$ is the probability the road segment is in the class, *i*, defined by the bound pairs *a* and *b* which are either [0, 1], [1, 2], or [2, 3]. $\sigma$ was chosen so that a $CQV_0$ in the middle of the two bounding *CQV*s for a given class class will contain exactly two standard deviations within that class, i.e., a 95% chance the road segment belongs to the class (see Fig 4 for an example).

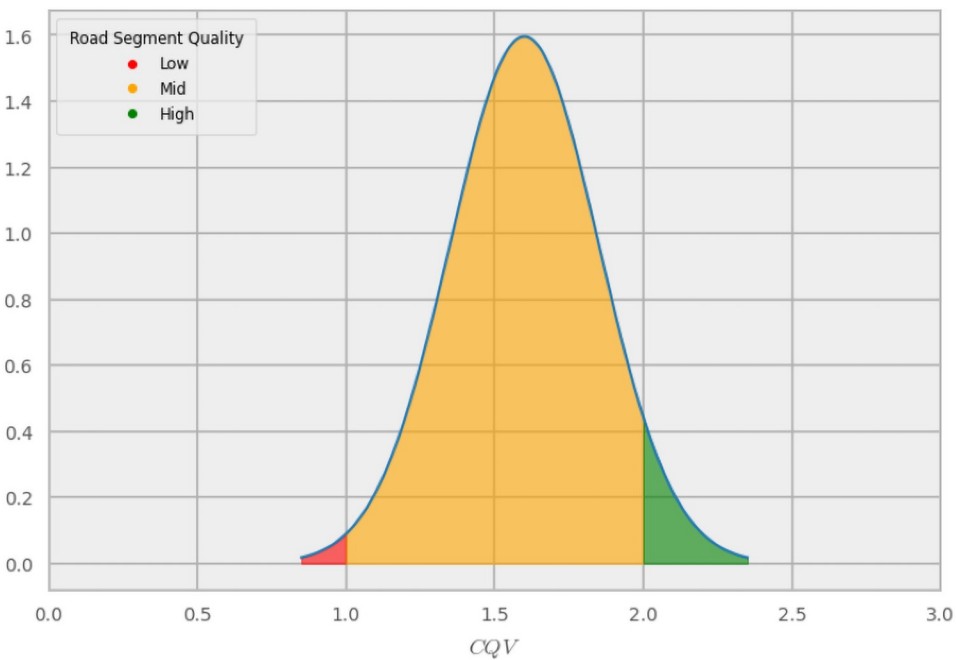

**Fig 4. Probability labeling.** Example of a $CQV$ of 1.60 indicating a road segment in the mid-quality class. The probabilities for the label were based on the proportion of the area under the curve of a normal distribution ($\sigma = 0.25$) that falls into each of the three classes. In the case of this road segment, the label is [0.008,0.937,0.055].

This equation results in a set of three probabilistic soft labels for each observation, one for each road class. These were used to train the network, with validation (and testing) data defined as one-hot-encoded lists.

For each of the pre-trained architectures tested, the top, i.e., the final 1000-class fully-connected layer, was removed. After the pre-trained model, the output was flattened and two fully-connected layers were added with a dropout layer in between. The first fully-connected layer contained 256 neurons with ReLU activation and the final layer utilized a softmax activation function outputting to three classes.

As part of the fine-tuning process, the network was first trained with the weights of the pre-trained models frozen to initialize the weights in the two added fully-connected layers, then the base model weights were unfrozen and trained again with a lower learning rate. Adam was chosen as the weight optimization algorithm and categorical cross entropy as the loss function. A grid search was initially used on a subset of the data to develop ideas for the optimal training batch size and learning rate hyperparameters. Depending on the model, a learning rate of 0.01 or 0.001 was used during the initialization training and $10^{-4}$ or $10^{-5}$ was used during fine-tuning. Batch sizes of 32 or 64 were used for the larger Virginia dataset and 16 for the Nigerian road dataset. Early stopping and model checkpoint callbacks were used during training to monitor validation loss, stop training when validation loss did not decrease after a specified number of epochs, and to save the model with the highest validation accuracy.

For testing on Nigerian roads, the highest performing base model was initially trained on a binary subset of the Virginia data—534 low-quality and 1,068 high quality for a total of 1,602 images and two classes. All 1,000 Nigerian roads were then tested. The Nigerian roads were labeled deterministically—[1, 0] hot-encoding for unpaved roads and [0, 1] for paved. For a final evaluation, 750 Nigerian roads were fine-tuned on the model at a very low learning rate ($10^{-5}$) and the model was tested on the remaining 250.

## Results & analysis

The results of each base architecture on the Virginia test set in order of overall accuracy are shown in Table 2.

There was an 8% improvement between the best and worst performing networks, InceptionResNet and Xception. No particular class correlated closely with overall performance. Each model performed relatively poorly with mid-quality roads, though performance between models varied most in this class. Inception was most successful in the class, correctly classifying 39% more mid-quality-labeled roads than the worst performer, Xception. With 47,136 high quality roads used in training, each model performed relatively well predicting in the class and there was less variance (9% total) between the models in the class. There was slightly better performance from smaller architectures versus larger; the top five models averaged 39,877,971 parameters versus 45,476,120 for the bottom four. Both variants of ResNet50 outperformed ResNet152 in our scenario. It was found that unfreezing only the last convolutional block for fine-tuning the VGG-based network produced better results by approximately 3% overall compared to unfreezing the entire base model. The effectiveness of the technique did not hold with the DenseNet-based network, but this may be due to the size, i.e. number of layers, of each convolutional block in DenseNet, or hyperparameter selection.

Hyperparameter choices strongly affected the relative performances of the models. A small number of hyperparameter configurations were tested on each network changing overall accuracy by as much as 6.5%. It was observed, in most cases, that using an ReLU activation function, as opposed to no activation, in the second-to-last fully-connected layer along with an order of magnitude higher learning rate during fine-tuning generated higher accuracies overall and on low and mid-quality roads at the expense of lower accuracies on high quality roads (DenseNet and Xception are two examples of exceptions that produced lower overall accuracies in this configuration). Performance within classes varied with hyperparamters as well for a given network. For example, adding ReLU activation to the second-to-last fully-connected layer and raising the fine-tuning learning rate an order of magnitude from $10^{-5}$ to $10^{-4}$ increased the Inception-based network's mid-quality performance from 35% to 60% (and its overall performance from second-worst to second-best).

It is worth noting the ResNet50 network pre-trained on BigEarthNet performed on par with the other models with no noticeable increase in time to convergence, demonstrating the original weights used in classifying land cover lend themselves well to identifying features in images of roads.

**Table 2. Architecture results.**

| Network Base Model | Overall (%) | Low Quality (%) | Mid-Quality (%) | High Quality (%) | # Parameters |
|---|---|---|---|---|---|
| InceptionResNetV2 | 78.0 | 79 | 58 | 89 | 60,629,219 |
| Inceptionv3 | 77.5 | 72 | 60 | 90 | 30,192,419 |
| VGG16 | 75.4 | 76 | 47 | 90 | 19,434,307 |
| ResNet50V2 | 75.3 | 64 | 56 | 91 | 49,255,939 |
| DenseNet201 | 73.4 | 59 | 44 | 97 | 27,169,859 |
| ResNet50 (BigEarth) | 72.8 | 63 | 55 | 88 | 24,159,214 |
| ResNet152V2 | 72.5 | 63 | 52 | 88 | 84,022,787 |
| Xception | 70.0 | 68 | 21 | 97 | 46,552,619 |

Results of the highest-performing run of each pre-trained architecture on the test data. The second column shows overall accuracy and the third, forth, and fifth columns show the accuracy on each class. The last column shows the total number of parameters in the network.

**Table 3. Ensemble results.**

| | | Prediction | | |
|---|---|---|---|---|
| | | Low | Mid | High |
| Label | Low | 166 | 47 | 2 |
| | Mid | 13 | 163 | 53 |
| | High | 3 | 55 | 363 |

Out of 215 examples of low quality roads, only 2 were predicted as high quality and only 3 of the 421 high quality roads were predicted as low.

The overall most accurate ensemble was a 3-member ensemble composed of the highest performing network in each class (IncpetionResNet, Inception, and DenseNet), achieving 80.0% classification accuracy, an improvement of 2.0% from the highest ranked individual model. A confusion matrix for the ensemble is shown in Table 3.

Compared to the averages of the three models, there was a 7% and 17% improvement in low and mid-quality road classification, respectively, but a decrease of 7% for high quality roads. The ensemble performed 11% better than the best performing individual model, Inception, at classifying mid-quality roads. The precision of low and high quality roads remained high with 87% and 91% of predictions for low quality and high quality being correct. For low and high quality roads, prediction to within one class (effectively top-2 accuracy) was 99.2%. The effectiveness of classifying the extreme cases was also confirmed in separate smaller tests on individual models, showing accuracies of up to 96% for datasets with only low and high quality roads. Again, with the ensemble, classification performance of mid-quality roads was still relatively inferior with an $F_1$-score of 66 and the ensemble predicting 23.1% of all mid-quality roads as high quality. In total, 97.1% of all incorrectly classified roads were off by only one class.

Randomly selected correctly classified images and randomly selected incorrectly classified images are shown in Fig 5. Errors in data collection (see Limitations), occlusion from trees, insufficient resolution, or off-centered or overly zoomed-in crops (see Limitations) may have contributed to many of the incorrect predictions. Using SHAP visualization (SHapley Additive exPlanations visualization technique and software developed at the University of Washington

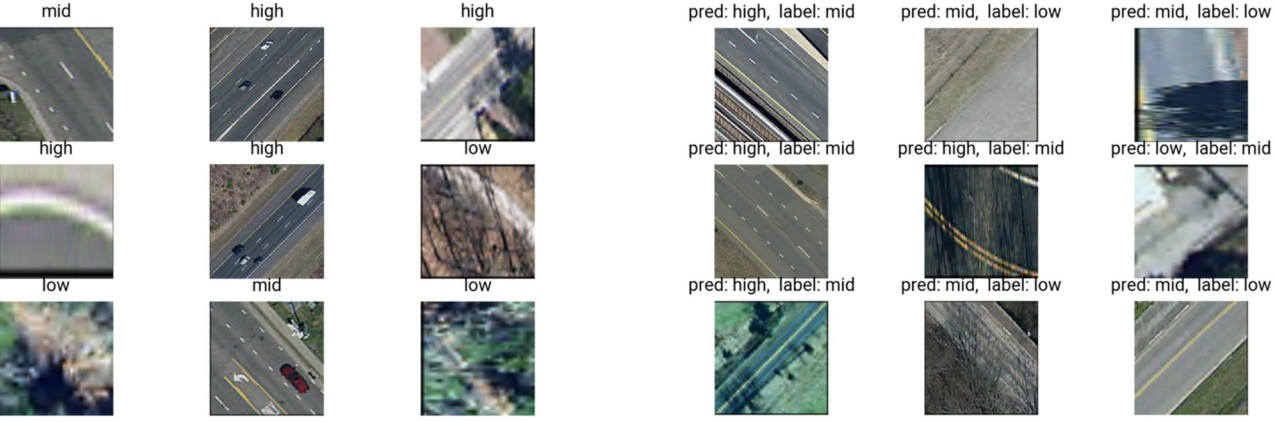

**Fig 5. Predictions.** On the left, a 3x3 group of correct predictions with associated labels and, on the right, a 3x3 group of incorrect predictions with predictions and labels.

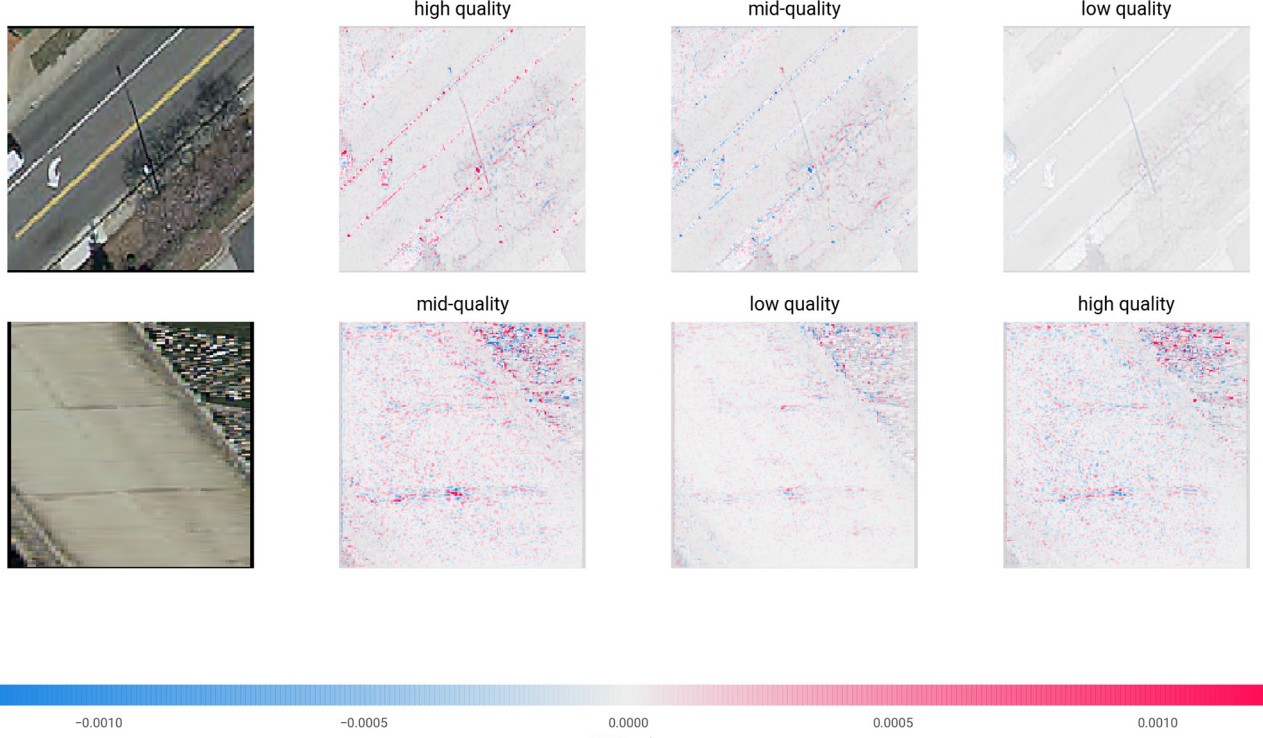

**Fig 6. Virginia prediction analysis.** The above shows two random correctly classified test images in the first column. The second, third, and forth columns show the pixels/features that contributed against and in favor of prediction for each of the three classes. For example, the upper-left image is a high quality-labeled road that was predicted by the model most likely to be a high quality road, then mid, then low. Blue pixels represent areas that work against classification in a given class and the red pixels represent areas that work for classification. In this case, the VGG16-based network was investigated.

Department of Computer Science [59]), Fig 6 shows an example of what factors played into the correct classification of two images in the VGG-based network.

For the image in the first row in Fig 6, the road lines and arrow played a large factor in its high quality designation and those same features prevented it from being classified as mid-quality. In the second image, the overall color of the road was an important factor for mid-quality classification, possibly along with the diagonal spaces between the concrete blocks that compose the road, and some shrubbery to the side.

For the binary evaluation of Nigerian roads, the InceptionResNet network trained on two classes performed at 67.3% on all Nigerian roads. Subsequent fine-tuning of the model on a 750-image subset of the roads raised the classification accuracy to 94% which is around that of binary classification on Virginia roads. Table 4 shows the resultant confusion matrix for the

**Table 4. Nigeria results.**

|  |  | Prediction | |
|---|---|---|---|
|  |  | Unpaved | Paved |
| Label | Unpaved | 115 | 12 |
|  | Paved | 3 | 120 |

Confusion matrix for the 250 Nigerian roads tested.

classification. SHAP evaluations confirm, in most cases, the fine-tuned model successfully distinguishing features (primarily color) of the actual roads to determine classification.

While this paper focused on the results of categorical classification, predictions of the continuous measurement of road quality, *CQV*, were performed with several models. Using the same dataset used in the classification ensemble, a 5-member stacked ensemble with linear regression resulted in a mean squared error (MSE) of 0.23 and a mean absolute error of 0.32, which is 10.6% of the full range of 3. The members of the ensemble, with lowest MSE first, were ResNet152, VGG16, InceptionResNet, Inception, and DenseNet201. No other models were tested for CQV prediction. The ensemble slightly improved on approximation from the best individual model, ResNet152, lowering the MSE from 0.27.

## Discussion

The work presented in this paper provides five contributions to the literature. First, we contribute to a growing body of evidence that contemporary orthographic imagery has sufficient resolution to capture variance in road quality across multiple geographies. While some initial work into this [17] had suggested the possibility, the work presented here provides a thorough exploration of different deep learning approaches in a wider range of geographic contexts. Second, and related to this, we illustrate that different model architectures have markedly different capabilities for this class of problem, suggesting the need for more tailored approaches to convolutional networks designed for satellite imagery analysis. Third, we introduce—to our knowledge—the first application of fuzzy-class-membership for object qualification using satellite data and convolutional neural networks. Fourth, we perform the first continuous estimate of road quality from satellite imagery. Finally, we provide the first example of the use of a phone app combined with machine learning for road quality prediction.

The relatively high proportion of low and mid-quality labeled roads predicted as one class higher may partly be owed to errors in data collection as inadvertent motion of the collection device will falsely raise the perceived roughness of a given road segment (see Limitations below). High quality labels are less likely contain significant error, because inherently in this study, the gradient of experimental error points in the direction of higher roughness values. In addition, many of the roads labeled mid-quality may be rougher portions of an overall high quality road, e.g., a segment containing potholes. In these instances, many of the features common in high quality roads such as dark color and defined lines will still be present, but the resolution of the satellite image will likely not be high enough to distinguish finer details like potholes. It is also important to note the temporal mismatch (between one and three years) between when the satellite images were captured and when the road quality data was collected on the ground. This issue has been noted in other satellite learning tasks, but is particularly significant in our study since road quality can change drastically over short periods (i.e., due to weather or construction).

Using soft labels added uncertainty to the experimentally collected data, which was particularly useful for roughness values that are close to class thresholds, whereas a hard label provided a lower level of precision with no uncertainty. Utilizing soft labels resulted in an observed accuracy improvement of about 2% for a given model.

Our independent analysis in an international setting showed modest ability for a model trained on high and low quality roads in Virginia, USA to classify paved and unpaved roads in Nigeria. Contrasted to the in-situ data collection approach of Cadamuro et al. in Kenya [17] (see Table 1), our transfer-learning approach performed 4 percentage points better in our binary Nigeria case study. This suggests that transfer-learning approaches may be able to achieve similar levels of accuracy to approaches reliant on in-situ data collection. Consideration

should be given to the fact that the model was not initially trained on the task of paved vs unpaved but rather relative roughness. Therefore, some Virginia images in the low quality class included paved roads conflicting with the evaluation attempt on Nigerian roads. Upon fine-tuning the model on some of the Nigerian roads, the model performance rose significantly (up 27%, to 94%). Misclassified roads tended to be intermediate cases, with area outside the road often influencing prediction more than the actual road, a downside of less exact cropping displaying a wider area.

## Limitations

Due to the human-centered experimental nature of collecting data via app, some amount of error was introduced into the data and, ultimately, the labeling of images. For example, while recording, a user may receive a message or phone call causing vibration or significant movement. Other movements can occur if the device is not properly secured in place or if the user, for example, lifts the phone to check if it is recording. So, there is no certainty every data point was collected free of inadvertent movement and, thus, the assigned labels are not absolutely indicative of ground truth. Even ground truth is ultimately subjective in the context of app-based road roughness collection since initialization requires diverse users to estimate class. As seen in Fig 5, some incorrect predictions for quality higher than the label may have been influenced by these errors in data collection as some of the roads do not look (to the human eye) to belong to the labeled class, such the bottom rightmost image. If the app is used again, to combat errors, future versions could pause recording when certain other apps receive notifications or when movement is detected beyond normal vibrations. For this study, some of these errors were accounted for as extreme vibratory readings and filtered downstream in processing. Using an app does have advantages in terms of scale, ease-of-use, deployability, cost, and coverage compared to traditional rolling or vehicle-attached road profilometers.

A second limitation is in the highly variable lengths of roads traveled over small time intervals—i.e., how long of a road to treat as the stretch along which a given roughness was recorded. With regard to satellite image cropping, the downside of using the min and max lat/longs to crop an image, as apposed to an arbitrary number of pixels from a central point in every direction, is that the resulting images are not squares, but rectangles. When resized to a square for uniform use with a model, if there is little variation in lat or long, the result may look like an out-of-proportion and stretched image. Approximately 10% of the images fall into this category. A finer but more computationally costly cropping method, where differences between minimum and maximum coordinates are checked and then adjusted, if necessary, would help alleviate this issue (however further evaluation on a binary test case was performed on cropped square images centered around a single coordinate pair yielding no improvement). Additionally, another 5–10% of cropped images are noticeably off-center due to inexactness in geolocation with respect to the roadway of interest. This appears as images where the road is not centered and contains a disproportionate amount of non-road area.

## Future directions

Aside from improvements in app design, image cropping techniques, and data collection practices, future studies could benefit in training and testing from more diverse data. A related challenge is that, due to the limited number of road types, spatially adjacent road patches are contained in both the train and test sets. Collecting on more roads would better allow us to hold-out consecutive segments of road (e.g., within 1 km) from appearing in both train and test sets. The geographic diversity in data collection can also be expanded for improved generalization to regions dissimilar to the mid-Atlantic U.S. in climate, foliage, development style,

and road design. The effectiveness of this idea was demonstrated by the improvement after fine-tuning with Nigerian roads. As with Virginia, effects local to Nigeria such as soil color and vegetation likely biased the model regionally, though one can imagine a deployed model consistently updating on small subsets of data from newly collected geographic areas becoming better and better at predicting road quality overall regardless of the global location and regardless of whether or not that particular region's roads have been included in prior fine-tuning. Data aside, other existing network varieties not tested in this study including recurrent neural networks, and further developments in CNN and Quantum CNN (QCNN) architectures may provide additional improvements to satellite image object qualification tasks. Fine-tuning networks pre-trained on satellite image datasets such as BigEarthNet may also prove valuable.

## Conclusion

In this paper, we sought to explore the capability of convolutional neural network architectures for identifying road quality from remotely sensed imagery. We integrated a novel Android application for collecting road quality information, high-resolution satellite imagery, and used these as inputs to test a variety of CNN architectures (Table 2). This approach achieved a top-1 accuracy of 80.0% with 99.4% of predictions falling within the actual or an adjacent class. There was some performance variance among the models depending on hyperparameter configuration and no single network performed better than any other on more than one class. An ensemble of the most accurate models in each class provided an overall improvement over any individual model, particularly with intermediate cases. The result of an estimation seeking to predict a continuous score of road quality ($CQV$) showcased the general ability of the methods to infer meaningful information from the satellite images to produce reasonable estimates of road quality. Finally, our exploratory test case on Nigerian roads showed the flexibility of this approach, with only a small amount of local data from Nigeria providing sufficient information to apply a transfer-learning based approach and achieve high levels of accuracy (94%) in the related task of detecting if roads are paved or unpaved.

Accurate and precise low-cost remote road assessment has the potential for effective use in several realms including targeting road repairs, international aid allocation, and vehicle routing. By monitoring the performance of construction firms and contractors, it can empower governments, donors, and policymakers to identify particularly hazardous roads, improving public safety and enabling better efficiency of public spending. With it, aid organizations can more readily monitor investments made to infrastructure in developing areas. Companies in the business of routing the public can use continuous and integrated forms of the technology to incorporate road quality into how they navigate customers or autonomous vehicles. Because of the ubiquitous availability of satellite imagery and the presence and importance of roadways in the lives of almost everyone in the world, this work should have near-term applications in many domains.

## Supporting information

**S1 Appendix. Technical innovation in the use of CNNs to analyze satellite imagery.**
(PDF)

## Acknowledgments

The authors acknowledge William & Mary Research Computing for providing computational resources and technical support that have contributed to the results reported within this paper. URL: https://www.wm.edu/it/rc.

The authors would also like to thank the following Commonwealth Cyber Initiative (CCI) fellows and affiliates for their help in collecting data and providing helpful insights: Taima Aliriani, Austin Anderson, Karim Bahgat, Calvin Bertoncini, Jiaying Chen, Genevieve Evins, Rini Gupta, Clare Heinbaugh, Sarah Larimer, Emilio Luz-Ricca, Linda Ma, Eric Nubbe, Yaw Ofori-Addae, Grace Smith, Jacob Somer, Anthony Stefanidis, Krishna Tejo, and Olivia Wachob of William & Mary; Sai Gurrapu, Minh Nguyen, and Charles Tan of Virginia Tech; Vi Nguyen and Matthew Wootten of George Mason University; Allan Brewer Pedin and Hannah Allen of Christopher Newport University; Chloe Adzima and Mian Shah of Virginia Commonwealth University; and Vadim Kudlay, Madison London, and Abdul Qadeer Rehan of the University of Richmond.

## Author Contributions

**Conceptualization:** Peter Kemper, Dan Runfola.

**Data curation:** Ethan Brewer, Jason Lin, John Hennin.

**Formal analysis:** Ethan Brewer, Jason Lin.

**Funding acquisition:** Peter Kemper, Dan Runfola.

**Methodology:** Ethan Brewer, Dan Runfola.

**Project administration:** Ethan Brewer, Dan Runfola.

**Software:** Ethan Brewer, Jason Lin.

**Supervision:** Ethan Brewer.

**Validation:** Ethan Brewer.

**Writing – original draft:** Ethan Brewer.

**Writing – review & editing:** Ethan Brewer, Dan Runfola.

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
