## [Decision Letter · Decision Letter 0]

24 May 2021

PONE-D-21-14133

Predicting Road Quality using High Resolution Satellite Imagery: A Transfer Learning Approach

PLOS ONE

Dear Dr. Brewer,

Thank you for submitting your manuscript to PLOS ONE. After careful consideration, we feel that it has merit but does not fully meet PLOS ONE’s publication criteria as it currently stands. Therefore, we invite you to submit a revised version of the manuscript that addresses the points raised during the review process.

ACADEMIC EDITOR:

Based on the comments received from the reviewers and my own observation, I suggest minor revisions for the article.

We look forward to receiving your revised manuscript.

Kind regards,

Thippa Reddy Gadekallu

Academic Editor

PLOS ONE

Journal Requirements:

3. Please ensure that you refer to Figure 3 in your text as, if accepted, production will need this reference to link the reader to the figure.

4. We note you have included a table to which you do not refer in the text of your manuscript. Please ensure that you refer to Table 3 in your text; if accepted, production will need this reference to link the reader to the Table.

5. We note that Figures 2, 3, 6, 7 and 8 in your submission contain satellite images which may be copyrighted. All PLOS content is published under the Creative Commons Attribution License (CC BY 4.0), which means that the manuscript, images, and Supporting Information files will be freely available online, and any third party is permitted to access, download, copy, distribute, and use these materials in any way, even commercially, with proper attribution. For these reasons, we cannot publish previously copyrighted maps or satellite images created using proprietary data, such as Google software (Google Maps, Street View, and Earth). For more information, see our copyright guidelines: http://journals.plos.org/plosone/s/licenses-and-copyright.

You may seek permission from the original copyright holder of Figures 2, 3, 6, 7 and 8 to publish the content specifically under the CC BY 4.0 license. 

If you are unable to obtain permission from the original copyright holder to publish these figures under the CC BY 4.0 license or if the copyright holder’s requirements are incompatible with the CC BY 4.0 license, please either i) remove the figure or ii) supply a replacement figure that complies with the CC BY 4.0 license. Please check copyright information on all replacement figures and update the figure caption with source information. If applicable, please specify in the figure caption text when a figure is similar but not identical to the original image and is therefore for illustrative purposes only.

Reviewers' comments:

Reviewer's Responses to Questions

**Comments to the Author**

1. Is the manuscript technically sound, and do the data support the conclusions?

Reviewer #1: Yes

Reviewer #2: Yes

2. Has the statistical analysis been performed appropriately and rigorously? 

Reviewer #1: Yes

Reviewer #2: Yes

3. Have the authors made all data underlying the findings in their manuscript fully available?

Reviewer #1: Yes

Reviewer #2: Yes

4. Is the manuscript presented in an intelligible fashion and written in standard English?

Reviewer #1: Yes

Reviewer #2: Yes

5. Review Comments to the Author

Reviewer #1: - Paper is well written. Author should add a little background of the study and limitations of the existing works and clearly explain the contributions at the end of the introduction.

- All the key terms of the equations must be mentioned

- Reorganize the introduction, trying to explain every word of the title.

- I can see some paragraph in introduction, related work and proposed approach which should be merged. Please reduce text and improve the representation of this section.

- Relevant literature review of latest similar research studies on the topic at hand must be discussed

- Authors should add the most recent reference:

1) Anomaly Detection in Automated Vehicles Using Multistage Attention-Based Convolutional Neural Network, IEEE Transactions on Intelligent Transportation Systems

2) CANintelliIDS: Detecting In-Vehicle Intrusion Attacks on a Controller Area Network using CNN and Attention-based GRU, IEEE Transactions on Network Science and Engineering

Reviewer #2: 1. What are the main contributions of the current work?

2. The 1st section can be split into introduction and related works.

3. Summarize the findings of the related works in the form of a table.

4. Some of the recent works on CNN and image processing applied in several applications such as the following can be discussed in the paper: "Hand gesture classification using a novel CNN-crow search algorithm, A novel PCA–whale optimization-based deep neural network model for classification of tomato plant diseases using GPU, Image-Based malware classification using ensemble of CNN architectures (IMCEC)".

5. Compare the current work with recent state-of-the-art.

6. PLOS authors have the option to publish the peer review history of their article (what does this mean?). If published, this will include your full peer review and any attached files.

Reviewer #1: No

Reviewer #2: No

---

## [Author Response · Author response to Decision Letter 0]

1 Jun 2021

Please see file "Response to Reviewers".

---

## [Decision Letter · Decision Letter 1]

4 Jun 2021

Predicting Road Quality using High Resolution Satellite Imagery: A Transfer Learning Approach

PONE-D-21-14133R1

Dear Dr. Brewer,

We’re pleased to inform you that your manuscript has been judged scientifically suitable for publication and will be formally accepted for publication once it meets all outstanding technical requirements.

Kind regards,

Thippa Reddy Gadekallu

Academic Editor

PLOS ONE

Additional Editor Comments (optional):

Reviewers' comments:

Reviewer's Responses to Questions

**Comments to the Author**

1. If the authors have adequately addressed your comments raised in a previous round of review and you feel that this manuscript is now acceptable for publication, you may indicate that here to bypass the “Comments to the Author” section, enter your conflict of interest statement in the “Confidential to Editor” section, and submit your "Accept" recommendation.

Reviewer #1: (No Response)

Reviewer #2: All comments have been addressed

2. Is the manuscript technically sound, and do the data support the conclusions?

Reviewer #1: Yes

Reviewer #2: Yes

3. Has the statistical analysis been performed appropriately and rigorously? 

Reviewer #1: Yes

Reviewer #2: Yes

4. Have the authors made all data underlying the findings in their manuscript fully available?

Reviewer #1: Yes

Reviewer #2: Yes

5. Is the manuscript presented in an intelligible fashion and written in standard English?

Reviewer #1: Yes

Reviewer #2: Yes

6. Review Comments to the Author

Reviewer #1: This should be a seprate paragrpah "Our paper is structured as follows. In \\Related 23

works", we review the relevant literature on the application of computer vision to 24

satellite imagery. In \\Data", we discuss our data collection methods, and \\Methodology" 25

provides the technical approach we test for road quality classification. We introduce our 26

results in the following section, and finally provide a discussion and conclusion in the 27

final two sections."

- The contributions should be precies and listed above the paper structure.

- most of the equations are not refered in the text.

- All tables should be same style like table 2 is with borders and table 3 without borders.

Reviewer #2: The authors have done a good job in addressing all the comments and suggestions. The paper is improved significantly and is in a good shape now. I recommend the paper to be accepted in the current form.

7. PLOS authors have the option to publish the peer review history of their article (what does this mean?). If published, this will include your full peer review and any attached files.

Reviewer #1: No

Reviewer #2: No

---

## [Editor Report · Acceptance letter]

28 Jun 2021

PONE-D-21-14133R1 

Predicting Road Quality using High Resolution Satellite Imagery: A Transfer Learning Approach 

Dear Dr. Brewer:

I'm pleased to inform you that your manuscript has been deemed suitable for publication in PLOS ONE. Congratulations! Your manuscript is now with our production department. 

Kind regards, 

on behalf of

Dr. Thippa Reddy Gadekallu 

Academic Editor

PLOS ONE